# Failure Analysis on a Collapsed Flat Cover of an Adjustable Ballast Tank Used in Deep-Sea Submersibles

**Fang Wang** [1], **Mian Wu** [2], **Genqi Tian** [3], **Zhe Jiang** [1], **Shun Zhang** [1], **Jian Zhang** [2] and **Weicheng Cui** [1,4,*]

[1]   Shanghai Engineering Research Center of Hadal Science and Technology, Shanghai Ocean University, Shanghai 201306, China; wangfang@shou.edu.cn (F.W.); zjiang@shou.edu.cn (Z.J.); Shun_zhang@rainbowfish11000.com.cn (S.Z.)

[2]   Jiangsu Provincial Key Laboratory of Advanced Manufacture and Process for Marine Mechanical Equipment, Jiangsu University of Science and Technology, Zhenjiang 212003, China; wumianmail@126.com (M.W.); zhjian127@163.com (J.Z.)

[3]   School of Materials Science and Engineering, Shanghai Jiao Tong University, Shanghai 200240, China; sjtutgq@sina.com

[4]   School of Engineering, Westlake University, Hangzhou 310024, China

*   Correspondence: wccui@shou.edu.cn

**Abstract:** A flat cover of an adjustable ballast tank made of high-strength maraging steel used in deep-sea submersibles collapsed during the loading process of external pressure in the high-pressure chamber. The pressure was high, which was the trigger of the collapse, but still considerably below the design limit of the adjustable ballast tank. The failure may have been caused by material properties that may be defective, the possible stress concentration resulting from design/processing, or inappropriate installation method. The present paper focuses on the visual inspections of the material inhomogeneity, ultimate cause of the collapse of the flat cover in pressure testing, and finite element analysis. Special attention is paid to the toughness characteristics of the material. The present study demonstrates the importance of material selection for engineering components based on the comprehensive properties of the materials.

**Keywords:** flat cover; adjustable ballast tank; fracture; high-strength maraging steel; toughness

## 1. Introduction

Deep-sea manned/unmanned submersibles are the necessary high-tech equipment for ocean exploration. It is used to carry crews or equipment to various deep-sea complex environments for efficient exploration, scientific investigation and resource exploitation [1,2]. Deep-sea submersibles contain multiple complex systems. Buoyancy regulation system is one of the important subsystems of the submersible. The adjustable ballast tank is one of the key components, which can ensure that the submersible has a good ability of depth setting and weight fine-tuning [3–5]. Through the high-pressure seawater pump, seawater is pumped in and out of the ballast tank, and the buoyancy balance of the submersible in seawater is adjusted within a certain range.

Reducing the weight for better performances such as range, speed, and payload is a significant consideration for the designers of the submersibles, as the weight-to-displacement ratio is used to evaluate the structural efficiency of underwater vehicles [1,2]. The weight of the submersible is distributed among the main components. At present, a spherical adjustable ballast tank with its promising application to a sea depth of 11,000 m and a volume of 300 L is designed, where domestic

ultra-high strength maraging steel 18Ni(350) was used for first time for this purpose, as shown in Figure 1. The adjustable ballast tank is composed of upper and lower hemispheres, which are connected by a Huff clamp. An O-ring seals the hemispheres at the end. Fine-tuning makes the weight of the submersible maintain positive buoyancy or produce enough negative buoyancy for safely sitting at bottom. A flat cover is beneficial for installation of different cabin piercing parts. Sealing is carried out through a radial O-ring and is evenly locked with the sphere by screws. The selection of maraging steel for the current ballast tank is exactly based on the principle of strength enhancement for lower weight. This is also the first time that this material has been used in deep-sea pressure vessels. Some of the candidate materials for underwater pressure hulls, such as titanium and high strength steel, and their main properties can be found in literature [6–9]. The candidate material for pressure hulls using 18Ni grade maraging steels has been preliminarily investigated in terms of their application history, performance, and manufacturing capability by the authors [9]. The damage tolerance related performances of 18Ni grade maraging steels, including yield ratio ($\sigma_y/\sigma_b$) and fracture toughness, have been evaluated. There are basically four wrought commercial maraging steels of the 18 percent nickel family i.e., 18Ni (200), 18Ni (250), 18Ni (300), and 18Ni (350) with yield strength ranging from 1400 MPa to 2400 MPa [10]. In the development of 'MIR' submersibles in the 1980s [11], new techniques to produce high strength, high nickel-content steel 18Ni(250) for pressure hulls are applied for its two pressure spheres. Up to now, other grades of 18Ni series of maraging steels have no application experience in deep-sea pressure hulls.

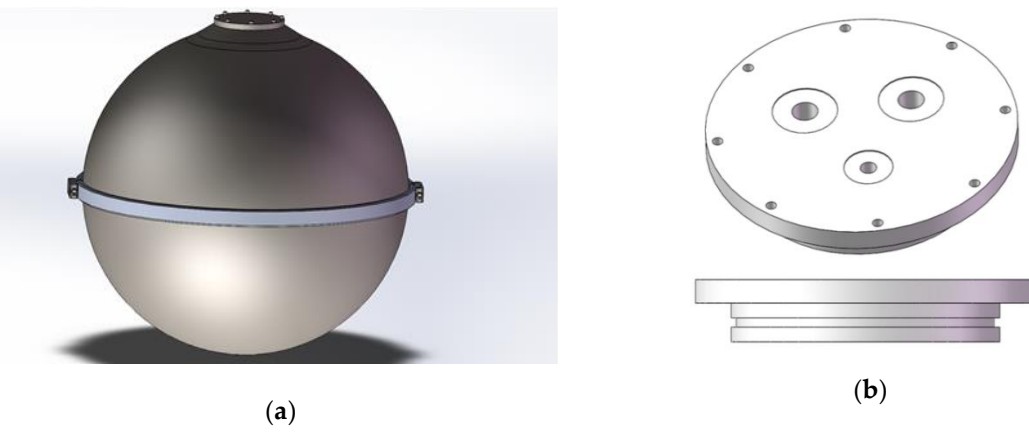

(**a**)　　　　　　　　　　　　　　　　　　　　　　　　　　　(**b**)

**Figure 1.** Sketch of the adjustable ballast tank and its sealing cover: (**a**) front view of the adjustable ballast tank; (**b**) description of the sealing cover.

The increase in strength is very powerful in reducing the weight of pressure hulls, but the only concern is that increased strength generally leads to a decrease in toughness [12,13] while it is generally not considered in the design of deep-sea compressive components. Plane strain fracture toughness is an important factor to represent the crack resistance property of the material. Wherein, 18Ni(250) with yield strength of 1700 MPa has the plane-strain fracture toughness level of 85–110 MPa·m$^{1/2}$, but the value of 18Ni(350) with yield strength of 2400 MPa rapidly reduces to 30–50 MPa·m$^{1/2}$ [9,12,13]. Improving the toughness and plasticity of maraging steel can be done in various ways, such as reducing harmful elements or gas content in steel by a double vacuum smelting process, controlling inclusions morphology, adjusting microstructure by special processing, and heat treatment processes [14]. Results have shown that when the harmful elements in 18Ni(350) maraging steel are reduced to $10^{-5}$ magnitude at the same time, the number and volume of inclusions are greatly reduced, which is an important reason for the remarkable improvement of fracture toughness of ultra-pure 18Ni(350) maraging steel. The influence of the toughness on the resistance to cracking for deep-sea components needs to be examined.

The hydraulic pressure test on the adjustable ballast tank with its sealing flat cover was carried out to examine its performance. The ultimate goal of pressure test was to check whether it can

endure the external pressure of 126.5 MPa (i.e., 115 MPa for 11,000 m deep-sea environments times a safety factor of 1.1 for pressure testing. Note, the actual pressure value for the 11,000 m deep-sea environment is 113.8 MPa and this value has been used in the design calculation, but here for the test, a more conservative value of 115 MPa is used). There are sensor mounting holes and high pressure pipe mounting holes on the sealing cover. However, the flat cover collapsed during the loading process of external pressure in the high-pressure chamber. The pressure was high, which was the trigger of the collapse, but still considerably below the design limit of the hull. The failure can be caused by unexpected defective material properties, the possible stress concentration resulting from design/processing, or inappropriate installation method. In this paper, the mechanical properties of the actual materials used in the collapsed flat cover are re-examined by sampling and testing of the broken parts. Non-metallic inclusions analysis, micro-structure analysis, and fracture surface analysis are conducted to acquire the possible fracture cause. Furthermore, finite element analysis based on fracture mechanics is conducted to understand the ultimate cause of destruction. The analysis results demonstrate the importance of material selection for engineering components based on the comprehensive properties of the materials.

## 2. Description of the Material and Structure

### 2.1. Chemical Composition and Material Properties

The preparation process of the material consists of vacuum induction furnace smelting and vacuum consumable furnace remelting. Solution heat treatment was conducted at 820 °C and kept for 1 h; aging treatment was conducted at 510 °C and kept for 4 h.

Chemical composition and mechanical properties of 18Ni (350) material are tested after sequential steps of melting, melt treatment practice, forging process, heat treatments, machining operations, and inspection, which is the same way as the cabin shell material was treated. Their chemical compositions and mechanical properties are given in Tables 1 and 2 and the stress–strain curve is depicted in Figure 2. This material has high tensile strength, the yield strength of which can reach about 2258 MPa and there is no obvious yield platform. However, its elongation is only 5.4%. The current design code for the spherical hull used in deep-sea submersibles and the design guideline for its flat cover have not specified the requirements for elongation of the material. The designers of the current spherical adjustable tank select 18Ni (350) instead of traditional material considering its high strength to reduce the weight of the tank, which will benefit the capability of the whole submersible.

**Table 1.** Chemical composition of 18Ni(350), (wt.%).

| Ni | Co | Mo | Ti | Al | C | Si | Mn | P | S |
|------|-------|------|------|-------|-------|------|------|-------|-------|
| 18.80 | 11.72 | 4.41 | 1.32 | 0.125 | 0.006 | 0.02 | 0.01 | 0.004 | 0.002 |

**Table 2.** Mechanical properties of 18Ni(350).

| $\sigma_y$, (MPa) | $\sigma_b$, (MPa) | *A*, (%) | *Z*, (%) | *E*, (MPa) | $v$ | *KV* at 20 °C, (J) |
|------|------|------|------|------|------|------|
| 2258 | 2324 | 5.4 | 31.0 | 188 | 0.3 | 13.5 |

where $\sigma_y$: Yield strength; $\sigma_b$: Ultimate tensile strength; *E*: Young's modulus; $v$: Poisson's ratio; *A* (%): Total elongation; *Z* (%): Area reduction.

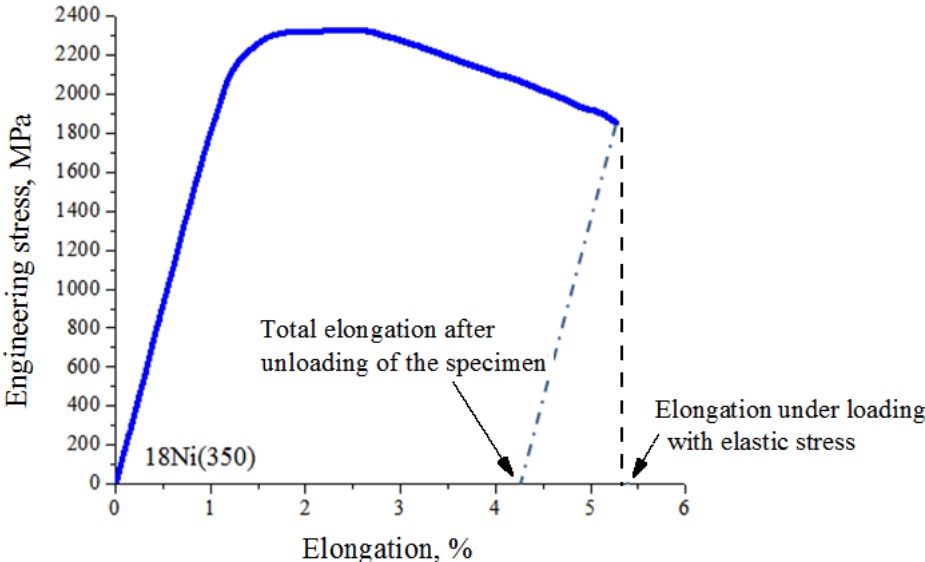

**Figure 2.** Stress–strain curve of 18Ni(350) (solution heat treatment: 820 °C for 1 h; aging treatment: 510 °C for 4 h).

### 2.2. Design and Geometry of the Flat Cover

The opening of the ballast tank is sealed by flange connection through the flat cover. The calculation of the thickness of the flat cover is carried out according to the requirements of the circular flat cover specified in the guidelines of steel pressure vessels [15].

$$\delta_p = D_c \sqrt{\frac{k\,P_c}{[\sigma]^t\phi}},\tag{1}$$

where $D_c$ = 176 mm is the calculated diameter of the flat cover; $k$ = 0.25 is the structural characteristic coefficient; $P_c$ is the calculated pressure, which can be 1.4 times the designed working pressure, i.e., 1.4 × 113 = 158.2 MPa; $[\sigma]^t = \sigma_y/1.5$. In order to make the strength safety factor larger, the yield stress applied by the designer presently is reduced to a lower level, 1900 MPa, then $[\sigma]^t$ = 1266.7 MPa; $\phi$ = 1.0 is the welding coefficient. This value of $\delta_p$ = 31 mm is obtained in calculation but was 32 mm in practice, including the thickness of 12 mm in the part of Φ190 mm and the thickness of 20 mm in the part of Φ150 mm, as shown in Figure 3. Necessary verification has been conducted by designers to check the strength and stiffness of the structure and it is considered to meet the requirements.

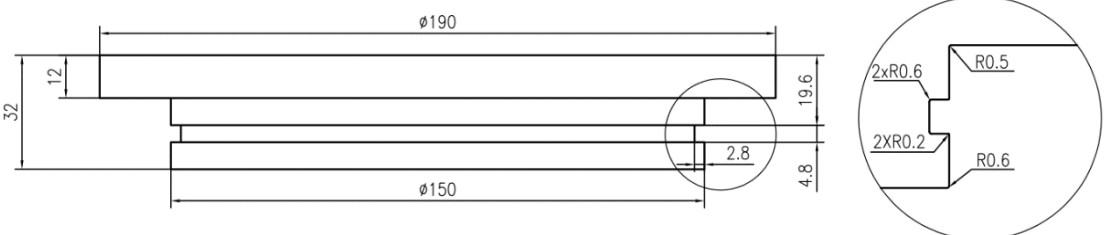

**Figure 3.** Dimension of the flat cover (unit: mm).

### 2.3. Failure of the Flat Cover During Testing

The spherical adjustable tank with its sealing cover was put into a high pressure chamber to examine its strength by pressure test before application. The pressure test was conducted in the high pressure chamber of the HAST laboratory at Shanghai Ocean University. There was no real-time video monitoring equipment in the pressure chamber and the test was carried out in several steps of cyclic

pressure in order to stop and examine it periodically. The scheduled loading scheme is shown in Figure 4. A shorter load cycle was done before formal test. The maximum pressure set for this cycle was 75 MPa. The ultimate goal of the pressure test was to check if the tank could endure the external pressure of 126.5 MPa (i.e., 115 MPa for the 11,000 m deep-sea environment times a safety factor of 1.1 for pressure testing). After test preparation, the tank with strain gauges pasted outside the shell in three typical positions was sent to the pressure chamber, then the first cycle for watertight and insulation tests combining with strain and stress measurements was conducted. The loading phase was carried out gradually with the step of pressure increase of 5 MPa and maintained for 3 min in the loading and unloading stage. It was presumed that when the first cycle was finished, the tank should be taken out from the pressure chamber for a further inspection. If there was no problem, the tank was sent to the chamber for a final test.

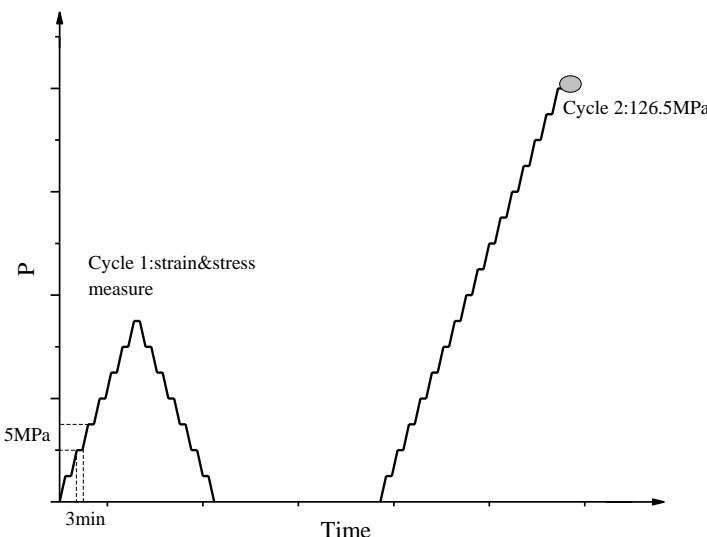

**Figure 4.** Loading scheme of pressure test.

However, when the tank was taken from the chamber after the first load cycle, it was found that the flat cover is cracked as shown in Figure 5. Local fracture occurred at the top part of the flat cover. The upper surface and fracture surface of the flat cover are respectively shown in Figure 5a,b. Based on the macroscopic observation, it was found that the flat cover bore large stress from outside to inside, and stress concentration occurred at the right-angle transition root. The crack morphology presented intergranular cracking, which was a brittle fracture.

Fracture occurred before the maximum pressure reached 75 MPa; however, it is difficult to assess at what pressure level the failure starts. The strain–load curves in Figure 6 show some abnormal phenomenon. Position 1 is at the opening reinforcement part of the spherical hull; position 2 is above the equatorial flange of the spherical hull; position 3 is at the bottom of the spherical hull. At a pressure of 40 MPa, there is a relatively obvious change of strain rate in the curves for three recording positions. This indicates that under this loading condition, the whole stress state of the spherical shell changes. At this point, the obvious stress concentration may appear on the flat cover. Because the external load is distributed both on the outer surface of the spherical shell and the flat cover, the change of stress state of the flat cover will cause stress concentration on the shell. From the fracture surface position, the stress concentration may come from the existence of crack-like defects, such as inadequate chamfer and scratches when the sharp part was processed. Therefore, in the next section, defects and fracture analysis will be carried out by means of microscopic observation and finite element calculation.

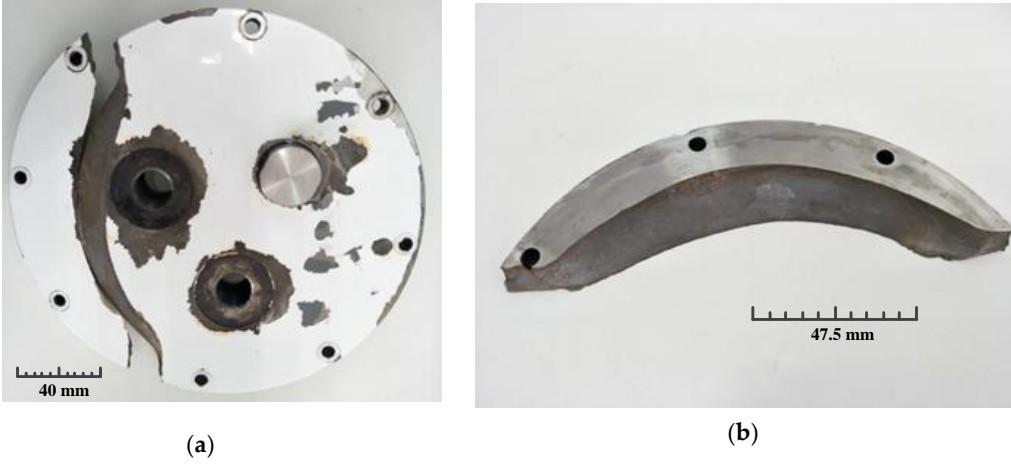

(**a**)                                    (**b**)

**Figure 5.** Photography of the cracked flat cover: (**a**) upper surface of the cover; (**b**) fracture surface in the outer edge of the cover.

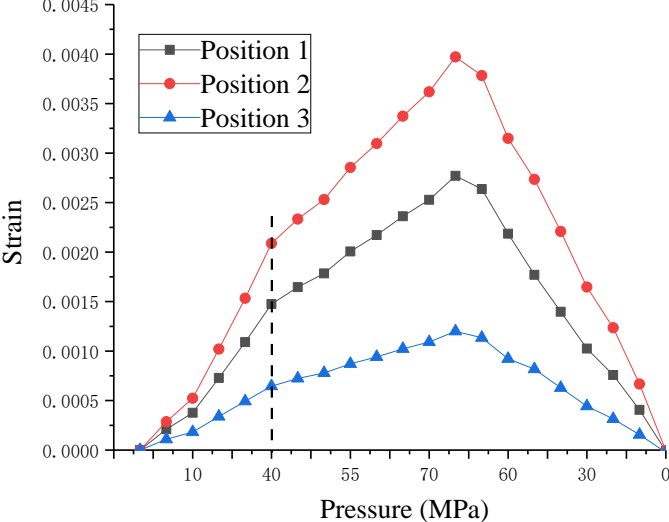

**Figure 6.** Recorded strains in the three positions outside the shell with variations of pressure.

## 3. Fracture Examination of the Flat Cover

Before the tank was manufactured, the material from the same furnace was tested to verify its conformance to the specified material standard and also the requirements of the designer. However, the same furnace material also had sampling uncertainty. Then in this section the tensile properties of the actual materials used in the flat cover were re-examined by sampling from the broken parts. The non-metal inclusions analysis, micro-structure analysis, and fracture surface analysis were conducted to find out the possible cracking cause.

### 3.1. Re-Examination of Tensile Properties

The static tensile test of a standard specimen with a diameter of 5 mm and gauge length of 30 mm [16] was carried out on SANS HST5106 hydraulic universal testing machine. Table 3 shows the results of the tests. It was found that the material has a higher tensile strength, yield strength, elongation, and reduction of area compared to the material presented in Table 2.

### 3.2. Re-Examination of Impact Properties

Standard Charpy V-notched impact specimens with length of 55 mm and square cross-sections of 10 mm × 10 mm [17] were used to determine impact toughness at room temperature on the PIT-752H

test machine. Table 4 presents the room temperature impact test results of the flat cover material. The fluctuation between the test values is small, and the average impact energy is 14 J. However, the material has low impact toughness. The minimum value of impact toughness of 18Ni(350) is 12 J while the impact toughness of 18Ni(250) is about 24 J, which may be one of the reasons for the cracking of the flat cover during the test.

**Table 3.** Re-examination results of tensile properties of flat cover material at room temperature.

| Specimen | $\sigma_y$ (MPa) | $\sigma_b$ (MPa) | A (%) | Z (%) | E (MPa) | $\upsilon$ |
|---|---|---|---|---|---|---|
| L1 | 2341 | 2421 | 6.4 | 47 | 186 | 0.300 |
| L2 | 2355 | 2423 | 6.6 | 42 | 188 | 0.310 |
| Average | 2348 | 2423 | 6.5 | 45 | 187 | 0.305 |

**Table 4.** Re-examination results of impact toughness of flat cover material at room temperature.

| Test Point | 1 | 2 | 3 | Average |
|---|---|---|---|---|
| KV, J | 12 | 16 | 14 | 14 |

### 3.3. Re-Examination of Hardness

Rockwell and Brinell Hardness (HRC) hardness test was carried out with Rockwell hardness tester. The load was 150 KGF with holding time of 10 s. The average hardness of the flat cover material was about 54 HRC.

### 3.4. Non-Metallic Inclusions Observation

Non-metallic inclusions in flat cover samples were inspected [18]. The results of non-metallic inclusions rating and the inclusion morphology in the most important part of the flat cover showed that there were fewer kinds of non-metallic inclusions. The inclusions specified in the standards include sulfide inclusions, alumina inclusions, silicate inclusions, and spherical oxides. The original state of the material meets the design requirements. Figure 7 shows the inclusion morphology of the sample.

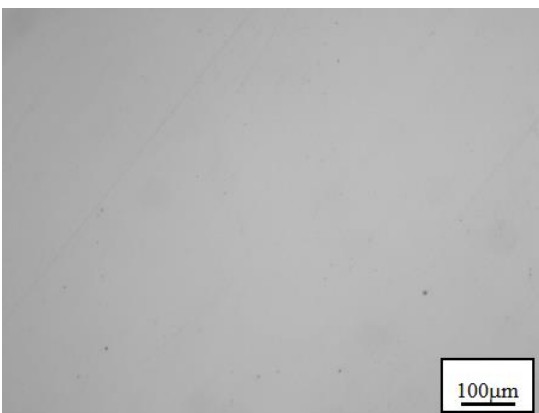

**Figure 7.** Inclusion morphology of the sample.

### 3.5. Microstructure Observation

Metallographic specimens were taken from the sealing cover. After mechanical grinding and polishing along the longitudinal section, the specimens were etched by an alcohol solution of $FeCl_3$ + hydrochloric acid at room temperature. The structure was observed under a ZEISS Axiovert 200MAT optical microscope (OM). The grading method of the grain size of samples was carried out according to grain size rating determination standard [19]. Figure 8 shows the structure of the flat cover material which is martensite.

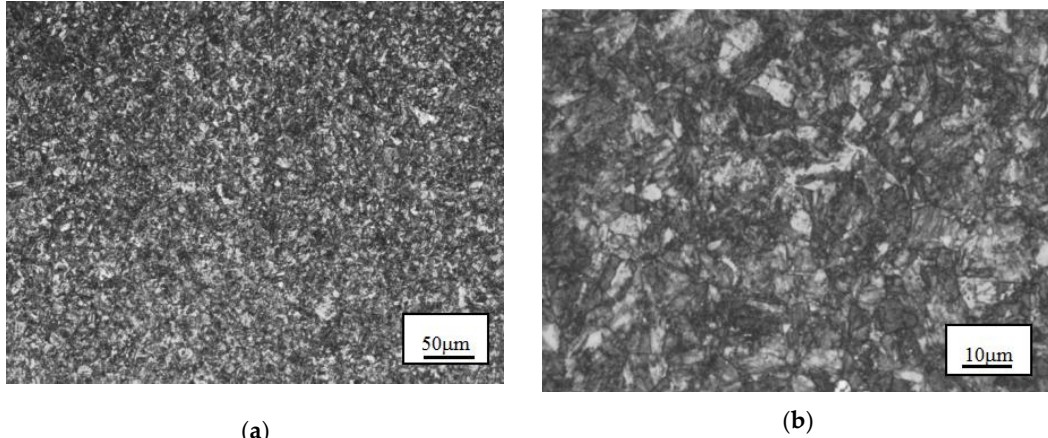

**Figure 8.** Metallographic structure of the cover sample: (**a**) low magnification; (**b**) high magnification.

Microstructure and morphology of the flat cover material were observed and analyzed by TESCAN VEGA3 XMU scanning electron microscopy (SEM). Figure 9 shows the SEM low and high magnification microstructures of the specimens. The martensite structure of the material was uniform, and a small amount of precipitated phase particles could be seen at the grain boundary.

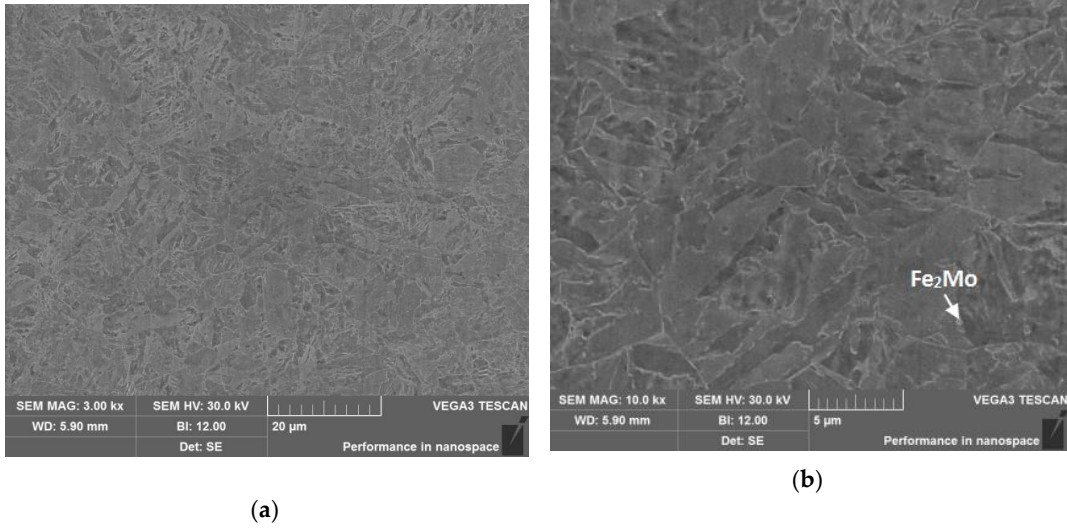

**Figure 9.** SEM microstructure of the cover sample: (**a**) low magnification; (**b**) high magnification.

Further observation was conducted under JEOL JSM-2100 transmission electron microscopy (TEM). Figure 10 shows the TEM morphology of the specimen. It can be seen that the width of martensite lath is about 0.2–0.8 µm, and a large number of precipitations were uniformly distributed in martensite matrix with high density. Electron diffraction analysis results show that a large number of rod-like particles are $Ni_3(Al, Ti)$-type $\gamma'$ precipitates, as shown in Figure 10b,c. In addition, large spherical granules in Figure 10c are $Fe_2Mo$ precipitates. Some studies [20] have shown that the precipitations of $Fe_2Mo$ occur with the increase of aging time and aging temperature.

*3.6. Fracture Surface Observation*

The fracture surface morphology was observed and analyzed by scanning electron microscopy after ultrasonic cleaning of the sample. Figure 11 shows the fracture morphology in the vicinity of the fracture source. It can be seen at the low magnification (Figure 11a,b) that the crack source is located at the root of the transition between the upper cover and the edge liner. It can be seen at high magnification (Figure 11c,d) that the crack source has an intergranular morphology and shows brittle

fracture. During the external pressure test, the flat cover was subjected to a large stress from the outside to the inside, and a large stress concentration was generated at the right-angle transition root in which a crack source was easily generated and resulted in fracture.

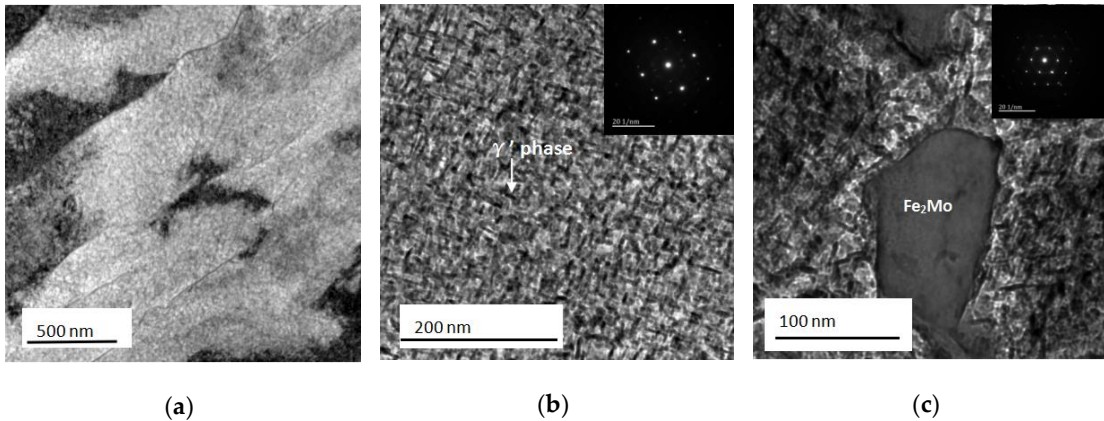

**Figure 10.** TEM microstructure of sealed top cover sample: (**a**) low magnification; (**b**) high magnification; (**c**) precipitated phase Fe$_2$Mo morphology.

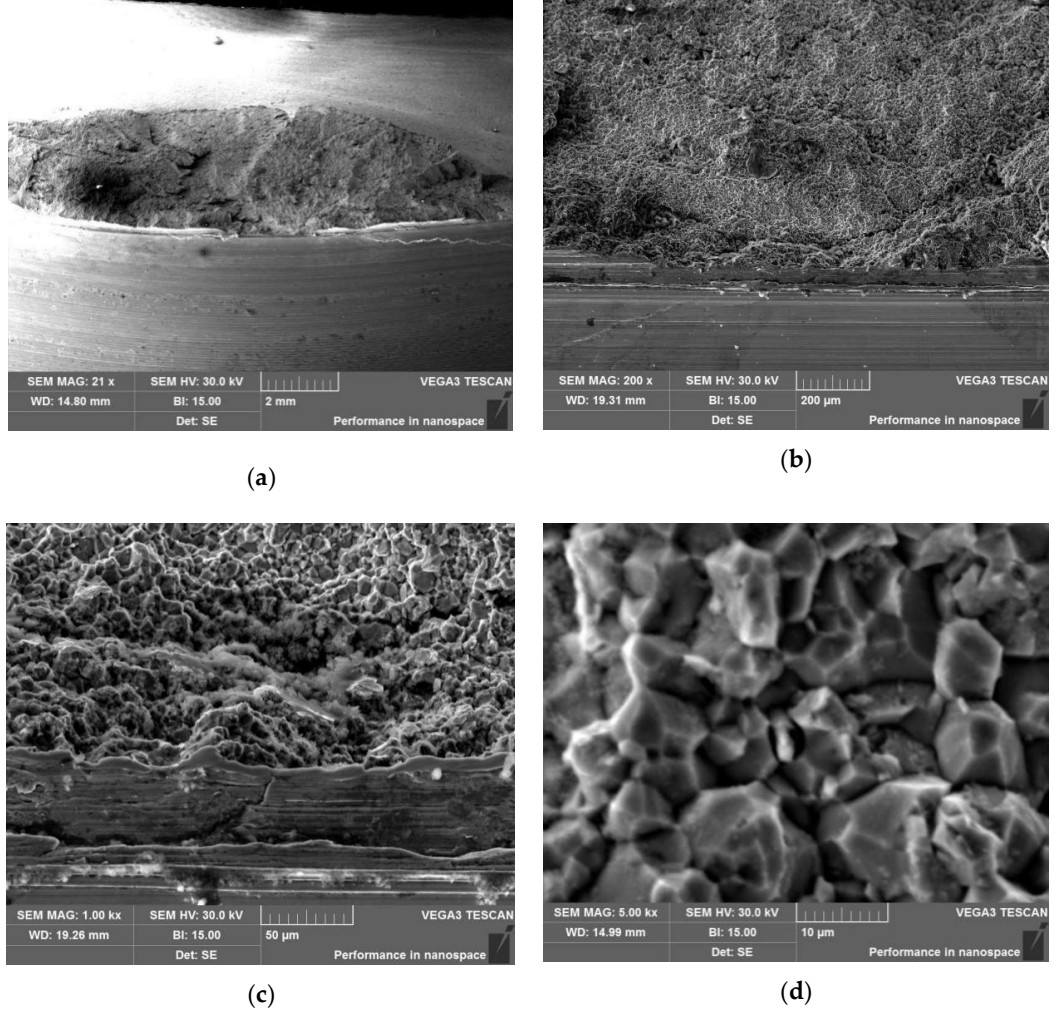

**Figure 11.** SEM morphology of crack source at the fracture surface of the flat cover: (**a**) low magnification morphology; (**b**) low magnification morphology at the crack source; (**c**) high magnification morphology at the crack source; (**d**) partially magnified morphology at the crack source.

Figure 12 shows the SEM topography of the tearing area of the flat cover. The tearing surface has a dimple morphology, which shows ductile fracture characteristics.

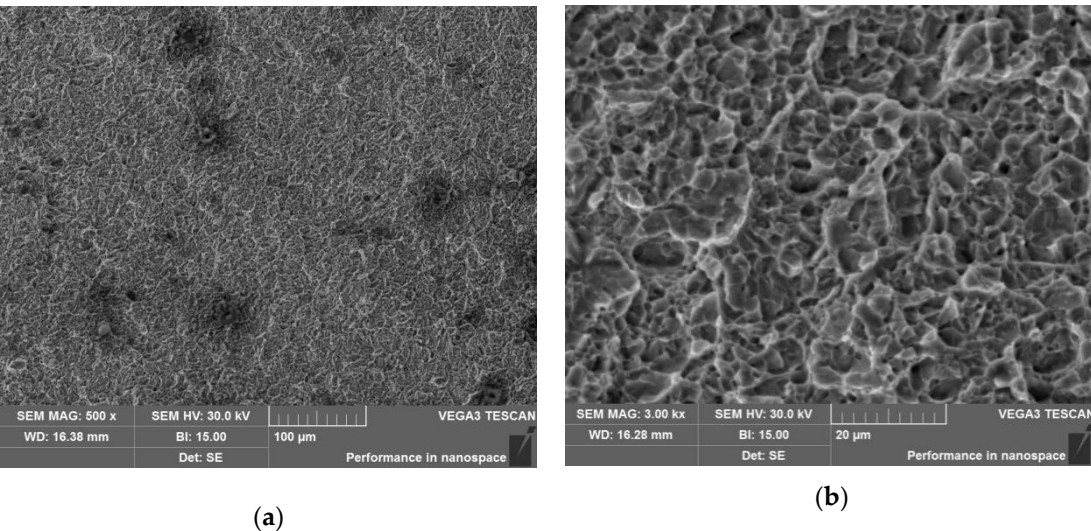

(**a**)                    (**b**)

**Figure 12.** Topography of cracking area of the flat cover: (**a**) low magnification morphology; (**b**) high magnification morphology.

## 4. Finite Element Analysis

Based on the observations in Section 3, it can be concluded that the final fracture of the flat cover is due to the large stress concentration generated at the right-angle transition root of the thicker central part and the thinner edge. A crack initiated from this point is shown in Figure 13. It was initiated from improper manufacture and has the length of 3 mm and the depth of 1.5 mm. Final failure easily occurs because the material has low impact and toughness properties. A presumed semi-elliptical surface crack is considered in this section to analyze the failure mode of the flat cover during loading process. The fracture criterion is that the stress intensity factor at the crack tip reaches the plane strain fracture toughness of the material.

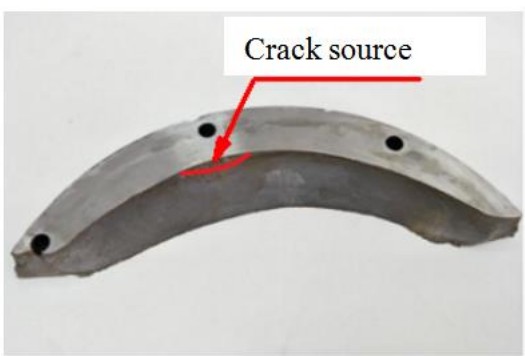

**Figure 13.** Crack position at the right-angle transition root.

### 4.1. Fracture Toughness of the Material

Fracture toughness tests are not conducted in the present study; however, it can be deduced from the widely accepted assumptions that the fracture toughness is taken to be a critical value of the maximum stress intensity factor for unstable crack growth [21]. According to this definition, the unstable crack growth will occur when $K_{max}$ reaches $K_{IC}$. In order to reduce test costs, this method was used to determine the fracture toughness of the material conveniently on the basis of the crack growth rate test for data over a wide range of growth rates [9] at four load ratios, i.e., 0.1, 0.3, 0.5, and 0.7.

The crack growth rate curves (*da/dN~ΔK* curves) at four load ratios of 0.1, 0.3, 0.5, and 0.7 were experimentally obtained in [9]. At each load ratio, two groups of tests were carried out. When the crack growth curve began to transit from stable growth stage to unstable growth stage, the requirement of $K_{max} = K_{IC}$ should be satisfied. Then a value of $K_{IC}$ can be calculated based on each crack growth rate curve. The average of these values can be taken as the final value of $K_{IC}$ as shown in Figure 14. The data of $\Delta K(1 - R)$ are within a relatively stable range, and the fracture toughness of 18Ni (350) can approach 37~40 MPa·m$^{1/2}$, which agrees well with the data range in literature [12,13] and can be applied for fracture analysis. The dotted line shows the average level of the calculated data of $K_{IC}$, which was about 38 MPa·m$^{1/2}$. It must be emphasized that the toughness of this material is greatly reduced compared to 18Ni (250), which is traditionally used in the pressure hulls of deep-sea manned submersible [9].

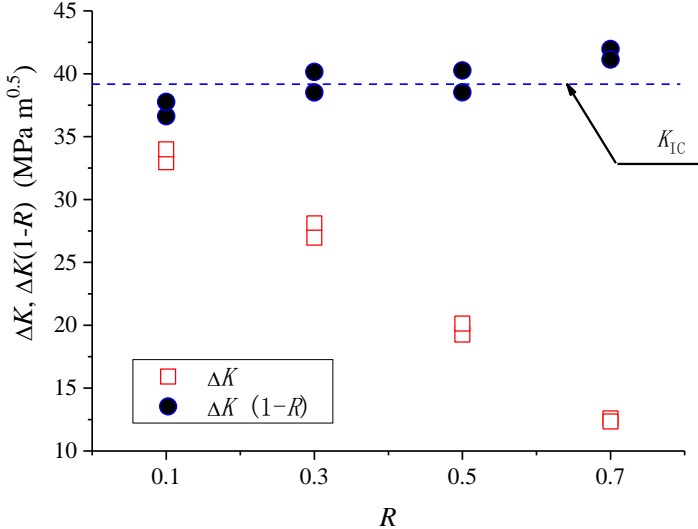

**Figure 14.** $\Delta K$ and $\Delta K(1 - R)$ values of 18Ni(350) determined by unstable crack growth properties.

### 4.2. Finite Element Analysis

SolidWorks software was used for three-dimensional modeling, and ABAQUS software is used for finite element analysis after modeling. C3D8R element is used for meshing. Local mesh encryption was made in the vicinity of key chamfers of the flat cover and contacts the part between the spherical model and the flat cover, with mesh size of 1 mm, compared to the mesh size of 20 mm in other parts. The small chamfer was not included in the calculation model, because when the crack was inserted into the model, the stress concentration at the crack tip would be the dominant factor, and the stress concentration at the small chamfer was no longer important. The whole structure model was used in finite element calculation to obtain the critical stress area and the surface crack was inserted into the local model after stress analysis. Figure 15 gives the whole model for stress analysis. The ballast tank used in the deep-sea submersible was subjected to uniformly distributed hydrostatic pressure, which was applied both in the spherical shell and the flat cover. In order to eliminate the overall rigid body displacement without hindering the relative deformation, the boundary conditions were set symmetrically with six displacement components constrained by the three-point support method recommended in [22]. The contact surface between the flat cover and the sphere was set as the friction contact with the friction coefficient of 0.1.

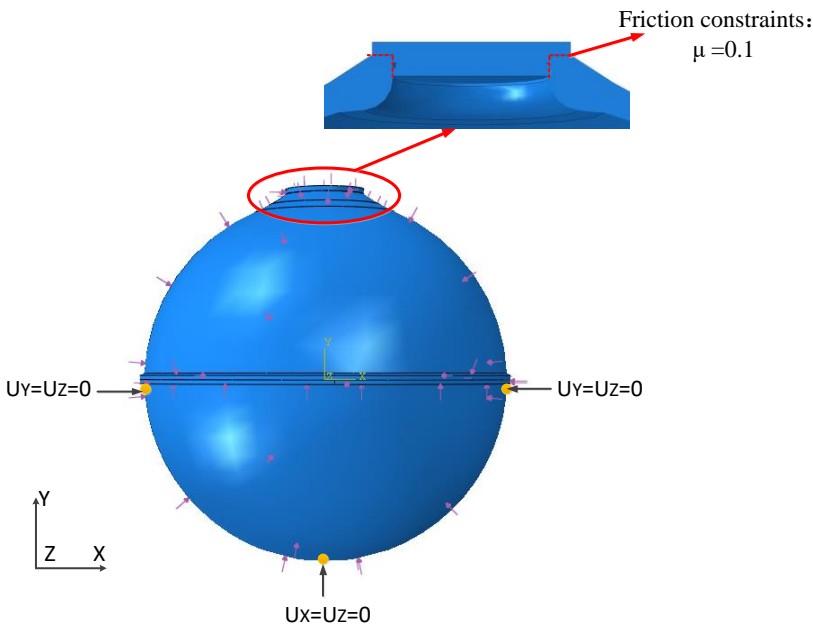

**Figure 15.** Finite element model for stress analysis.

When a crack is simulated, the crack will be inserted into a sub-model. The sub-model was taken out separately for the study. The calculation can be conducted by using the connection between the global model and the sub-model, thus greatly reducing unnecessary computation and time costs. According to the results of previous failure observations, the defective part of the cover is treated as a sub-model and the crack with the length of 3 mm and the depth of 1.5 mm is inserted into the model. After static analysis, the stress intensity factor of crack front can be obtained. The position relationship between the sub-model and the global model is shown in Figure 16.

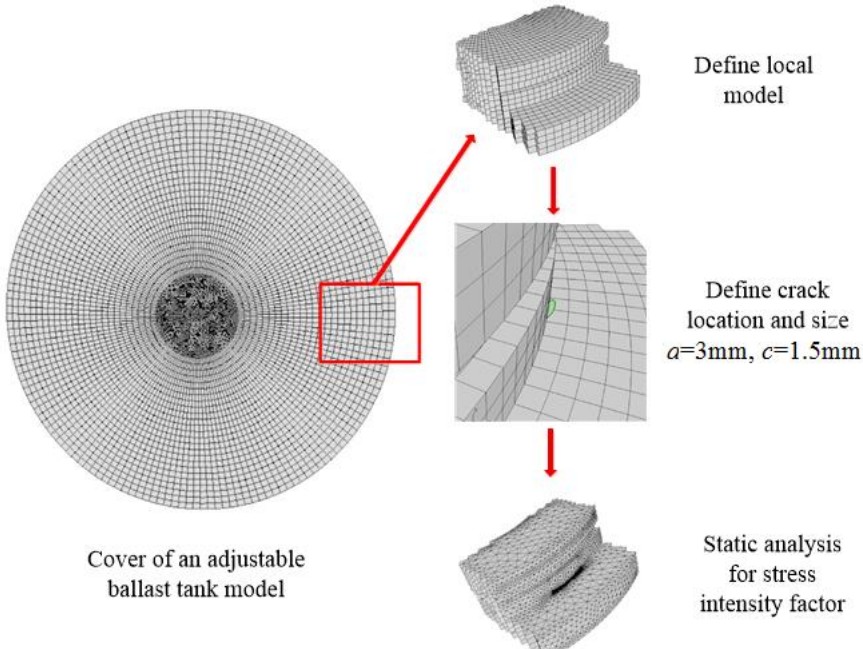

**Figure 16.** The position relationship between the sub-model and the global model.

When the flat cover is subjected to different external pressures from 5 to 85 MPa, the stress intensity factors of the crack can be obtained. Figure 17 depicts the stress intensity factors along the crack front with increasing the external pressure. Figure 18 shows the changing of the stress intensity factors at

the surface points and the deepest point with the external pressure. The stress intensity factor in the length direction is larger than that in the depth direction. The maximum value of the stress intensity factor occurs at the surface points. When the external load increases to about 75–85 MPa, the stress intensity factor at the crack tip reaches 35–42 MPa·m$^{1/2}$, i.e., approximating to the value of fracture toughness, 38 MPa·m$^{1/2}$, shown in Figure 14, which is the ultimate cause of the damage.

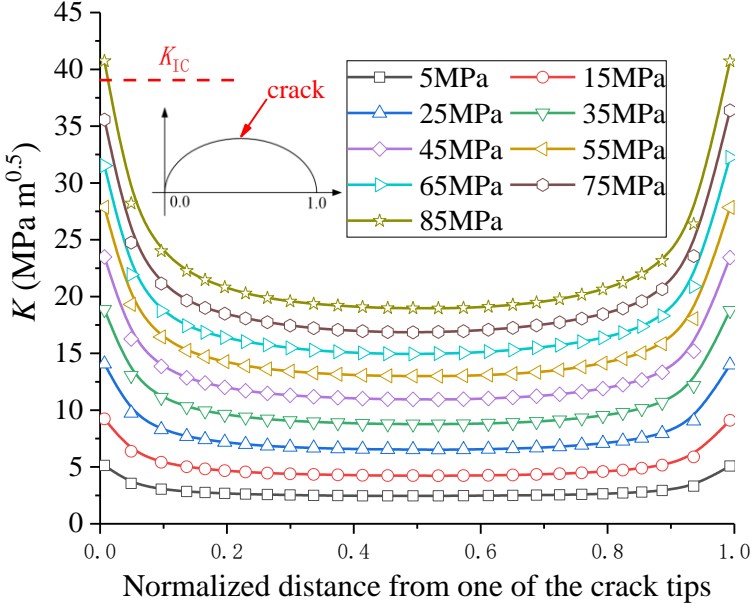

**Figure 17.** Stress intensity factor along crack front with increasing external pressure.

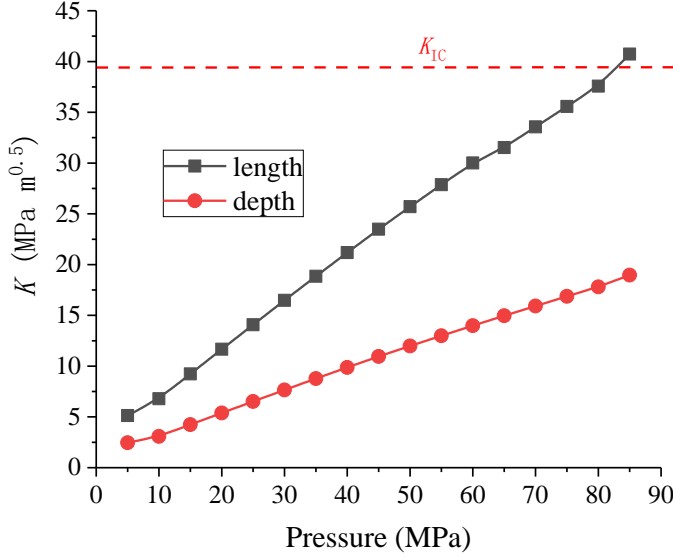

**Figure 18.** Stress intensity factor at the surface points and the deepest point.

## 5. Summary and Conclusions

In this paper, a destroyed flat cover made of 18Ni (350) used in an adjustable ballast tank during high pressure testing was analyzed. To find out the cause of the damage, the mechanical properties of the material used in the flat cover were re-examined by sampling and testing of the broken parts. Non-metallic inclusions analysis, microstructure analysis, and fracture surface analysis were conducted to find out the possible fracture cause. Finite element analysis based on fracture mechanics to understand the ultimate cause of destruction was conducted. The following conclusions and recommendations can be made:

(1) Maraging steels are expected to still have high toughness when improving their ultra-high strength level. 18Ni (250) grade is currently an acceptable candidate material for deep-sea pressure hulls, which has achieved satisfactory application experience. To obtain better weight-to-displacement ratio, the designers applied 18Ni (350) for an adjustable ballast tanks used in a deep-sea submersible. The strength of 18Ni (350) is higher than that of 18Ni (250), however its toughness is lower. In the current design guideline for deep-sea pressure hulls, the requirements for material toughness have not been emphasized. The trial of applying 18Ni (350) was made in the present study but unexpected failure of the flat cover occurred when load was relatively small.

(2) Non-metal inclusions analysis, microstructure analysis, and fracture surface analysis showed that the original state of the material meets the design requirements with fewer kinds of non-metallic inclusions. The material has low impact toughness, which is one of the reasons for the cracking of the flat cover during the test.

(3) From the design point of view, it is suggested to optimize the transition from right angle to chamfer at variable cross-sections, and to increase the thickness of the outer edge liner of the cover appropriately, which will reduce the possibility of damage. During processing, the surface finish of structural parts should be ensured, especially the root surface of the variable cross-section, so as to avoid surface processing defects.

(4) The reliability of deep-sea pressure hulls, especially the pressure hulls of manned submersibles, guarantees the safety of personnel and equipment. There must be attention paid to the comprehensive performance of materials. From the point of view of material selection, the application of 18Ni (350) maraging steel to the pressured structure in deep-sea environments is a risky choice. On the other hand, the 18Ni series of maraging steels have broad application prospects in deep-sea environments, so improving the toughness and plasticity together with increasing the strength level will be an important research direction [23].

**Author Contributions:** The individual contributions are specified as follows: conceptualization, F.W. and W.C.; methodology, F.W. and J.Z.; software, M.W.; validation, M.W., S.Z., G.T., and Z.J.; formal analysis, F.W.; investigation, S.Z.; resources, Z.J.; data curation, M.W. and G.T.; writing—original draft preparation, F.W.; writing—review and editing, W.C.; visualization, F.W.; supervision, W.C.; project administration, F.W. and W.C.; funding acquisition, F.W. and W.C.

**Funding:** This research was funded by the General Program of National Natural Science Foundation of China, grant number 51679133, the State Key Program of National Natural Science of China, grant number 51439004, and the "Construction of a Leading Innovation Team" project by the Hangzhou Municipal government.

**Conflicts of Interest:** The authors declare no conflict of interest.

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
