# Peer review of "Failure Analysis on a Collapsed Flat Cover of an Adjustable Ballast Tank Used in Deep-Sea Submersibles"

_applsci, doi:10.3390/app9235258_

Round 1

Reviewer 1 Report

in-depth discussion on the microstructure-material properties-failure mechanism relationships is needed.   The finite element analysis was presented in this paper. However, the information provided is quite limited. The software used, the mathematical model, boundary conditions, etc. are not clearly introduced. Normally the simulation needs to be conducted at the first place, then verified by the experimental result. It seems odd to do the simulation based on the experimental results. what is the final conclusion of the failure? Is it due to the improper design, machining defects or low impact toughness of the material?   The conclusions need to be revised to clearly state the findings.

Reviewer 2 Report

Dear authors,

The article you sent to publication is very interesting and the subject of the high-strength steels is still important and actual. The maraging steel is the structural material of extremely high strength but the improving the toughness of the steel is significant problem and subject of investigation of many research centres.

Your results are very interesting, especially from the practical point of view. In the attached file I marked in the red suggestions, comments and questions. In my opinion the article will be more clear and comprehensible for readers. Correction and complement of the article will change its style and make the paper more scientific. At present article's style exhibits the good "defectology" expertise.

I think my suggestions will not be heavy for you.

Best regards

Reviewer 3 Report

In this manuscript, the authors investigated the fractured flat cover as a part of ballast tank to find out the causes of the facture, where mechanical property tests, microstructural analysis and finite element analysis were used. Even though the manuscript seems to provide valuable information, it should be improved, especially for material characterizations. And also, I strongly recommend the authors should have English proofreading for clear explanation.

Here are some examples the authors should be considered for English proofreading.

On page 1, line 35, “Reducing the structural weight for better performance by increasing the achievable range/speed/payload is a significant…” -> for better performances such as range, speed and payload

On page 2, line 40, “a volume of 300 L is designed using domestic ultra-high strength maraging steel 18Ni(350)” -> is designed, where domestic ultra-high strength maraging steel 18Ni(350) was used.

On page 2, line 74, “if the structure satisfy the strength requirement” -> if the structure satisfies the strength requirements

In the summary and conclusion section, miss numbering (1)

And, here are some issues the authors should address for further consideration.

1. On page 4, line 161, the authors mentioned “ the elongation and section shrinkage of the material are low as expected comparing to Table 2. But, the results in Table 3 show better elongation and ductility property than those of the original material. Please check this sentence or the values in the Tables.

2. The authors did the impact test to see toughness characteristic of the material but only mentioned that the value of the toughness is too low. Please provide more information like the toughness characteristic gets to be deteriorated or if the low toughness is an intrinsic characteristic, why do the authors use the material because it can be easily expected that the failure occurs during service time.

3. In the section 3.4, the authors argued there are fewer kinds of non-metallic inclusions even the original material meets the design requirements. What are the design requirements and I wonder that the non-metallic inclusion were generated or they already existed in the material.

4. Usually, it is not easy to measure the grain size of a martensite phase due to its complex microstructure. Does the value of the grain size rating, 7.5 represent the grain for the martensite or that for the prior austenite grain?

5. The explanation on the fracture surface observation seems to be not clear to me. Please mark “crack source area” and explain why this area show the brittle characteristic even though the other fractured area has the dimple structure.

Round 2

Reviewer 1 Report

The authors have revised the manuscript based on the comments given by the reviewer. No further comments from the reviewer.

Author Response

Reviewer 1 accepted the revised version and no further comments.

Reviewer 2 Report

Dear Authors,

In attached file I marked my comments and questions. The explanation of obtained examination results should be extended.

It is evident that editing of English language and style is required.

Best regards

Reviewer 3 Report

The authors answered all issues I gave so the manuscript could be accepted for the publication.

Author Response

Reviewer 3 accepted the revised version and made no further comments.

Round 3

Reviewer 2 Report

Dear authors,

Please find attached file with my last suggestions and questions.

In my opinion English language should be editing in style to make your paper more clear. 

Best regards

Author Response

Thanks for the comments and recommendations. The revisions and replies have been made in the attached file according to the recommendations, comments and questions which are marked in the manuscript by the reviewer.
